

# A cross-sectional survey of avian influenza knowledge among poultry farmworkers in Indonesia

Saifur Rehman[1,2], Aamir Shehzad[1], Lisa Dyah Andriyani[4],
Mustofa Helmi Effendi[2], Zain Ul Abadeen[5],
Muhammad Ilyas Khan[3] and Muhammad Bilal[3,6]

[1] Laboratory of Virology and Immunology Division of Microbiology, Faculty of Veterinary Medicine, Airlangga University, Surabaya, East Java, Indonesia
[2] Division of Veterinary Public Health Faculty of Veterinary Medicine, Airlangga University, Surabaya, East Java, Indonesia
[3] Department of Epidemiology and Public Health, Faculty of Veterinary Science, University of Veterinary and Animal Sciences, Lahore, Punjab, Pakistan
[4] Food and Agriculture department Batu, Batu, East Java, Indonesia
[5] Department of Pathology Faculty of Veterinary Science, University of Agriculture Faisalabad, Faisalabad, Punjab, Pakistan
[6] Current affiliation: Faculty of Veterinary Medical Sciences, University of Calgary, Alberta, Canada

Corresponding author
Mustofa Helmi Effendi, mhelmieffendi@gmail.com

## ABSTRACT

**Background.** Avian influenza (AI) poses a serious threat to global public health, especially the highly pathogenic form. Awareness and protective behavior among the public, particularly the high-risk populations, are essential for prevention and control. This study aimed to ascertain the level of AI knowledge among Indonesia's poultry farmworkers.

**Methods.** This was a cross-sectional study conducted online. A predesigned standardized questionnaire, containing six demographic questions and 14 questions on AI knowledge, was used. The questionnaire was distributed *via* WhatsApp and email platforms. Volunteers (respondents) included 119 men and 81 women, aged 18–50 years, who work on poultry farms in Indonesia. Data were analyzed using the chi-squared and Fisher exact tests.

**Results.** The study's findings revealed that more than two-thirds (67.0%) of the respondents had heard about AI. Their primary sources of information were health workers (36.0%) and media, especially television (34.0%). The majority of the participants (91.3%) had good knowledge about AI as a contagious infection, transmissible from birds to other birds, animals, or humans. A total of 76.8% of the respondents believed that poultry workers and veterinarians were at high risk of contracting AI infection.

**Conclusions.** The study concluded that poultry workers had good knowledge about AI infection, transmission, and risk variables. Health workers and television were the main sources of information on AI. The level of AI knowledge was high among the respondents.

## INTRODUCTION

Avian influenza (AI), commonly known as "bird flu", is a highly contagious viral infection belonging to the family Orthomyxoviridae. It has the potential to infect both birds and humans. The strains of this virus can present in various ways in terms of severity, depending on their virulence (*OIE, 2020*). The first case of the highly pathogenic AI H5N1 (HPAI H5N1) strain in a human was recorded in Hong Kong in 1997 (*Yuen et al., 1998*) and live bird markets were thought to have contributed to this outbreak (*WHO, 2007*). Individuals who engage in the poultry industry or who interact directly with poultry may be more susceptible to AI than the general public, and thus may function as an AI transmission route to the general population (*Huang et al., 2015*). According to a report published by the World Health Organization (*WHO, 2017*) on March 16, 2017, 858 documented cases have resulted in 453 deaths in 16 countries since 2003. The human mortality rate from this disease in developing countries seems to change over time. In China, it was 100% in 2003 and then decreased to 50% by 2010. In Egypt, the human mortality rate peaked at 56% in 2003 and then dipped to 45% by 2010. Since 2005, HPAI H5N1 has been found in numerous other Asian countries, including Afghanistan, Bangladesh, India, Myanmar, Pakistan, and most recently, Bhutan and Nepal (*Timilsina & Mahat, 2018*). The HPAI H5N1 subtype has been endemic to poultry since 2003 in Indonesia and continues to cause significant social and economic losses for both the poultry industry and backyard farms (*Sumiarto & Arifin , 2008*; *WHO, 2011a*; *WHO, 2011b*). Poultry producers and the industry are suffering significant social and economic consequences (*Basuno, Yusdja & Ilham, 2010*; *Rushton et al., 2005*). Indonesia has the highest HPAI H5N1 human death rate in the world. Since the first outbreak in August 2003–May 2015, 199 (human) AI cases were confirmed in Indonesia using laboratory testing, of which 165 were fatal. Cases have been documented in Bali, Sulawesi, Sumatra, Lombok, and Java Island, with a majority being recorded in Java Island (*WHO, 2015*; *Kurscheid et al., 2015*). The Indonesian government has taken several measures to avoid the highly pathogenic AI (HPAI) virus, resulting in a decrease in disease outbreaks in poultry since 2012 (*FAO, 2012*) and a significant decline in human H5N1 infections since 2013 (*WHO, 2017*). *Morris et al. (2005)* identified several factors either directly or indirectly associated with the spread of the HPAI virus throughout Asia. These factors were unsafe handling and farming systems and activities, including the rearing of different poultry species in a free-range environment in rural or urban locations, using contaminated vehicles and bird cages to transport live birds, and the insufficiency or absence of biosecurity practices at live bird markets. Given these observations, AI knowledge among poultry farmworkers and other poultry industry stakeholders is vital for AI prevention and control in poultry and humans (*Rehman et al., 2022a*; *Rehman et al., 2022b*; *Rehman et al., 2022c*).

Human infections have been associated with the handling of dead or sick poultry in H5N1-affected areas, indicating that H5N1 illness in humans is spread primarily through infected birds (*Neupane et al., 2012*).

Data on the importance of knowledge in relation to the influenza pandemic have been less convincing. Although some studies have discovered the benefits of protective

behaviors (*Eastwood et al., 2009*; *Liao et al., 2011*), others have not (*Van der Weerd et al., 2011*). A previous study conducted by *MacMahon et al. (2008)* stated that poultry workers who are exposed to infected birds, poultry products, virus-contaminated objects, or environments have an occupational risk of infection with these viruses. Moreover, poultry workers at risk of AI virus exposure were found to include those operating in various poultry production systems or sectors, such as poultry farmers and their staff (*Leonard, 2009*). However, these studies found that poultry farmworkers and veterinarians are the most susceptible to AI infections if they are exposed to infected birds or virus-contaminated environments or materials. Several epidemiological studies have been published to assess the H5N1 infection risk factors in humans, especially when there is contact with poultry and poultry products (*Zhou et al., 2009*; *Van Kerkhove et al., 2011*; *Van Kerkhove, 2013*). AI virus exposure has been linked to contact with contaminated poultry blood, bodily fluids during food preparation, and working with poultry in markets or farms (*Radwan et al., 2011*). According to survey findings from the capital city of Kathmandu, Nepal, 38.7% of the country's butchers had some understanding of AI. However, none of the respondents showed sufficient knowledge or proper behavior (*Paudel, Acharya & Adhikari, 2013*). Previous studies among poultry farmworkers in Italy, Nigeria, and China revealed that HPAI knowledge was considerably higher among those with educational attainment and those who were perceived as being more susceptible to this infection (*Abbate et al., 2006*; *Fasina et al., 2009*; *Yu et al., 2013*). An earlier study conducted in Indonesia among small-scale poultry farmers indicated that those with a greater understanding of HPAI symptoms are more likely to implement good practices of poultry and poultry product handling and are more concerned about disease transmission risks (*Tiongco et al., 2011*). Moreover, urban poultry workers and consumers appear to be more knowledgeable about HPAI than their rural counterparts (*Barennes et al., 2007*; *Fasina et al., 2009*). These findings are not surprising considering that poultry workers and dealers have poor educational levels (*Alders et al., 2009*). In fact, there are no sufficient acceptable facilities for poultry workers to prevent AI infection in certain countries. Poultry workers and traders are not involved in disease control and surveillance programs, which are usually conducted by government organizations (*Alders et al., 2009*; *Azhar et al., 2010*).

The impacts of HPAI, information sources, and education initiatives (*e.g.*, mass media, training, and community mobilization activities) on the poultry workers or villagers' knowledge have been explored in some countries (*Azhar et al., 2010*; *Barennes et al., 2007*; *Kurscheid et al., 2015*; *Manabe et al., 2011*; *Neupane et al., 2012*; *Yu et al., 2013*). The primary HPAI information source in Nigeria, Laos, and Vietnam was television (TV; *Barennes et al., 2007*; *Fasina et al., 2009*; *Manabe et al., 2011*), whereas the radio was more essential in Nepal (*Neupane et al., 2012*). Similarly, previous studies conducted in Indonesia revealed that TV is the primary source of AI information in mass media (*Tiongco et al., 2011*; *Kurscheid et al., 2015*). Good public awareness and knowledge about specific diseases or infections are critical for their prevention and for successful outbreak control (*Dishman, Stallknecht & Cole, 2010*; *Van Nhu et al., 2020*; *Rehman et al., 2022a*; *Rehman et al., 2022b*; *Rehman et al., 2022c*).

In light of the above, this study's main objectives were to assess the knowledge level among Indonesian poultry farmworkers regarding AI and to identify the factors related to knowledge, such as sociodemographic traits and media usage. This study's findings are expected to help policymakers enhance AI knowledge and awareness among poultry farmworkers through educational initiatives (seminars and workshops).

## MATERIALS & METHODS

### Ethical considerations

The Animal Care and Use Committee, Faculty of Veterinary Medicine, University of Airlangga, Surabaya, reviewed and approved this study's protocol under approval letter number 2.KE.096.07.2021. Participants were given verbal information about the study's aims, purpose, and structure, as well as assurances of confidentiality.

### Study area

This cross-sectional study was conducted in five different provinces (Banten, Jawa Barat, Jawa Tengah, Jawa Timur, and Lampung) of Indonesia (Fig. 1).

### Study population

This was a descriptive, cross-sectional online survey conducted by using a predesigned questionnaire, targeting Indonesian poultry farms workers in the different provinces (Banten, Jawa Barat, Jawa Tengah, Jawa Timur, and Lampung; Fig. 1). The selected provinces were located on Java Island, which represents 60% of the human and 70% of the poultry (layer, broiler, breeder, and backyard) populations of Indonesia (*Sumiarto & Arifin , 2008*). This island was more affected by AI infection than other Indonesian islands, because of the high poultry and human populations.

The majority of respondents (125; 62.5%) were from the region's East Java province, which is characterized by a high-density poultry and human population. The inclusion criteria were all people (employees or farm owners) working at large-scale commercial poultry farms (broiler, breeder, and layer) and backyard poultry farms (raising a large number of poultry). The questionnaire was initially written in English but was translated into the region's native language (Bahasa Indonesia), to improve the response accuracy, reduce error margin, and avoid confusion among respondents. The questionnaire was created using Google Forms, which could be accessed by clicking on a link. It was distributed by the investigators *via* social media, such as WhatsApp, and electronic media, such as email platforms, from August 11, 2021 to October 10, 2021. The study's aim was concisely explained to each study participant prior to obtaining informed consent and filling out the study questionnaire.

The Raosoft online calculator was used to calculate the sample size (*Raosoft, 2004*), which is specifically intended for population surveys to calculate the sample size and determine how many replies are required to achieve the desired confidence level with a margin of error (usually 5%). This calculator is strongly recommended in such a study while considering the population size. The overall number of poultry farmworkers in Indonesia was estimated to be around 12 million (*Ferlito & Respatiadi, 2018*). The precise population

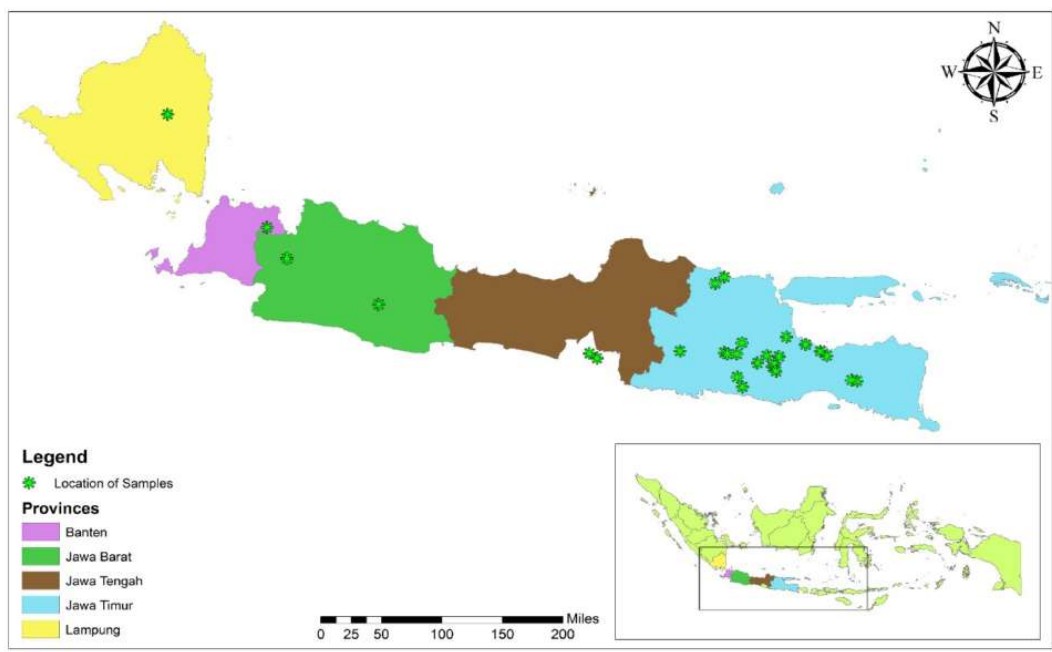

**Figure 1**   **Sampling area of the survey.**

of poultry farmworkers in the designated study areas was unknown, but an informed estimate of approximately 20,000 poultry farmworkers was obtained. The questionnaire was distributed among 450 people, to obtain a good response rate.

The authors used numerous WhatsApp groups associated with local veterinarians to identify commercial and backyard poultry farms with existing contact information so that the study questionnaire could be sent. In this context, it is important to note that small-scale poultry production farms (kept by households using family labor) were excluded from this study. The investigators kept track of the veterinarians' groups and reminded them regularly through WhatsApp and email, based on the respondents' convenience. The authors intended to reach as many outlets as possible to obtain reasonable and sufficient responses. Therefore, an online questionnaire was forwarded to approximately 450 participants to minimize the chances of error and maximize the response accuracy. Among these, 210 respondents filled out the forms. Only 10 questionnaires were determined to be missing important information, which were excluded from the final analysis (44.4% response rate). Only those respondents who completely answered all the questions in the study questionnaire were included.

## Questionnaire validation

A pilot study was conducted in the aforementioned provinces for two reasons: to ensure that the questionnaires were comprehensive and to ensure that the respondents were willing to participate in the study. The questions were written in English and Indonesian Bahasa. The survey was broadly distributed after correcting any errors and responding to minor suggestions concerning the language of the questions.

## Data collection tools

The data collection tools were adopted from previously published questionnaires from a study on Italian poultry workers and some modifications were made to align the questions with the local situation (*Abbate et al., 2006*),  as well as the WHO fact sheet on AI (*WHO, 2011a; WHO, 2011b*). The questionnaire comprised 20 items and was divided into two sections. The first part comprised six questions that investigated demographic variables and general information, including gender, age, residence, religion, educational level, and working status. There were 14 multiple-choice questions in the second section, with "yes"/"no"/"don't know" options. The question "Have you heard of avian influenza?" (yes/no) was used to assess public awareness of the disease; the question on sources of information, with options such as radio, TV, newspapers, health workers, and friends, was used to estimate the main sources of AI-related information among the participants. Furthermore, the participants were asked about the mode of transmission and vehicles of transmission, with "yes"/"no"/"don't know" options. A question on whether certain professional groups, such as poultry workers, butchers, or veterinary doctors, were at risk of contracting AI infection was used to assess professional risk discrimination ("yes"/"no"/"don't know").

## Data management

The knowledge scores were graded as follows: one for "yes" (positive) and zero for "no" and "don't know" (negative). The "don't know" response was merged with the "no" option because "don't know" is regarded a negative response. These scores were then converted into categorical variables: high (scores >80%), moderate (50%–80%), and low (<50%) (*Islam et al., 2017*).

## Data analyses

The acquired data were imported into the Statistical Package for the Social Sciences (SPSS), version 25.0 by the primary author. The typing errors were discovered and rectified. Given the nature of the study, descriptive statistics were conducted. Pearson's chi-squared ($X^2$) test or Fisher's exact test (if applicable) was used to analyze the relationship between different variables, with $p \leq 0.05$ being considered statistically significant.

# RESULTS

A total of 200 farmworkers from the five provinces of Banten ($n = 15$), Jawa Barat ($n = 15$), Jawa Tengah ($n = 30$), Jawa Timur ($n = 125$), and Lampung ($n = 15$) participated in this study (Fig. 1). The overall response rate was 44.4%.

## Sociodemographic background

Both male and female farmers who worked on poultry farms were considered in this study. Most of the respondents (59.5%, $n = 119$) were males, whereas 40.5% ($n = 81$) were females. The 31–50-year age range comprised 74.0% ($n = 148$) of the respondents, 25.0% ($n = 50$) were in the 20–30-year age range, whereas a small proportion of 1.0% ($n = 2$) were under 20 years of age. Of the 200 respondents, 55.5% ($n = 111$) resided in rural areas

**Table 1  Demographic characteristics of study respondents.**

| Variables | Characteristics | Frequency (*n*) | Percentage (%) |
|---|---|---|---|
| Gender | Male | 119 | 59.5 |
| | Female | 81 | 40.5 |
| Age | 18 years | 2 | 1.0 |
| | 20–30 year | 50 | 25.0 |
| | 31–50 year | 148 | 74.0 |
| Residence | Urban | 89 | 44.5 |
| | Rural | 111 | 55.5 |
| Religion | Muslim | 187 | 93.5 |
| | Christian | 5 | 2.5 |
| | Hindu | 2 | 1.0 |
| | Catholic | 6 | 3.0 |
| Educational Status | Non-primary | 5 | 2.5 |
| | Higher than primary | 195 | 97.5 |
| Working Status | Farm owner | 82 | 41.0 |
| | Paid employees | 118 | 59.0 |

and 93.5% (*n* = 187) practiced Islam. A majority of the respondents (97.5%, *n* = 195) had completed primary school, whereas only 2.5% (*n* = 5) did not. Additionally, more than half (59%, *n* = 118) of the participants were paid employees in the poultry business (Table 1).

## Awareness and sources of information on AI
Of the 200 respondents, 67.0% (*n* = 134) had heard about AI from various sources, including health workers (36.0%), TV (34.0%), friends (14.5%), and newspapers (14.0%). Only 1.5% had learned about it from the radio (Table 2).

## Mode of transmission
A high percentage (83.5%) of the participants were aware that AI was a contagious infection that affects all birds. The majority (95.0%) believed that AI was transmissible from animal to animal, whereas only 67.5% believed that it was transmissible from animal to human, an indication of its zoonotic nature. A small proportion (20.5%) stated that it could be transmitted from human to human. In addition to this, 50.0% of the participants stated that touching uncooked poultry and eggs could contribute to spreading AI. A total of 95% claimed that poultry, whereas 91% alleged that other birds were the main AI transmission sources (Table 2).

## Risk groups
An average of 76.7% of the respondents thought that poultry workers and veterinarians were more likely to contract AI infection than butchers. Overall, the study's findings demonstrated that the participants had strong knowledge dealing with AI infections (Table 2).

**Table 2 Knowledge of the participants regarding AI.**

| Sources of AI | Poultry farm workers $n = 200$ | | |
| --- | --- | --- | --- |
| | Responses | Number | % |
| Awareness and information sources | | | |
| Have you heard about AI? | Yes | 134 | 67.0 |
| | No | 66 | 33.0 |
| | Radio | 3 | 1.5 |
| | TV | 68 | 34.0 |
| Sources of information | Newspapers | 28 | 14.0 |
| | Health workers | 72 | 36.0 |
| | Friends | 29 | 14.5 |
| Is AI a contagious infection that affects all birds? | Yes | 167 | 83.5 |
| | No | 24 | 12.0 |
| | Don't know | 9 | 4.5 |
| Mode of transmission | | | |
| Animal to animal | Yes | 191 | 95.5 |
| | No | 4 | 2.0 |
| | Don't know | 5 | 2.5 |
| Animal to human | Yes | 135 | 67.5 |
| | No | 49 | 24.5 |
| | Don't know | 16 | 8.0 |
| Human to human | Yes | 41 | 20.5 |
| | No | 127 | 63.5 |
| | Don't know | 32 | 16.0 |
| Touching uncooked poultry | Yes | 102 | 51.0 |
| | No | 80 | 40.0 |
| | Don't know | 18 | 9.0 |
| Touching uncooked eggs | Yes | 77 | 38.5 |
| | No | 103 | 51.5 |
| | Don't know | 20 | 10.0 |
| Vehicles of transmission | | | |
| Poultry | Yes | 190 | 95.0 |
| | No | 3 | 1.5 |
| | Don't know | 7 | 3.5 |
| Birds (other than poultry) | Yes | 182 | 91.0 |
| | No | 4 | 2.0 |
| | Don't know | 14 | 7.0 |
| Other animals | Yes | 74 | 37.0 |
| | No | 87 | 43.5 |
| | Don't know | 39 | 19.5 |

**Table 2** (*continued*)

| Sources of AI | Poultry farm workers $n = 200$ | | |
|---|---|---|---|
| | Responses | Number | % |
| Risk groups | | | |
| Poultry workers | Yes | 159 | 79.0 |
| | No | 28 | 14.0 |
| | Don't know | 13 | 6.5 |
| Butchers | Yes | 100 | 50.0 |
| | No | 83 | 41.5 |
| | Don't know | 17 | 8.5 |
| Veterinarians | Yes | 149 | 74.5 |
| | No | 37 | 18.5 |
| | Don't know | 14 | 7.0 |

**Table 3 Relationship between awareness and demographic characteristics.**

| | Have you heard before about avian influenza? | | | *P*-value | 95% CI |
|---|---|---|---|---|---|
| Demographic | Characteristics | Yes<br>Frequency % | No | | |
| Gender | Male | 82 (69) | 37 (31) | 0.487 | 0.585–1.290 |
| | Female | 52 (64) | 29 (36) | | |
| Age | 18 years | 2 (100) | 0 (0) | 0.592 | N/A |
| | 20–30 year | 34 (68) | 16 (32) | | |
| | 31–50 year | 98 (66) | 50 (34) | | |
| Residence | Urban | 60 (67) | 29 (33) | 0.516 | 0.656–1.456 |
| | Rural | 74(67) | 37 (33) | | |
| Religion | Muslim | 124 (66) | 63 (34) | 0.704 | N/A |
| | Christian | 4 (80) | 1 (20) | | |
| | Hindu | 2 (100) | 0 (0) | | |
| | Catholic | 4 (66) | 2 (34) | | |
| Educational Status | Non-to-Primary | 3 (60) | 2 (40) | 0.534 | 0.409–3.633 |
| | Higher than Primary | 131 (67) | 64 (33) | | |
| Working Status | Farm owner | 56 (68) | 26 (32) | 0.433 | 0.624–1.403 |
| | Paid employees | 78 (66) | 40 (34) | | |

**Notes.**
CI, Confidence Interval; N/A, Not applicable.

## Awareness and demographic characteristics

Table 3 depicts the association between participant awareness and demographic characteristics. None of the variables had a significant relationship with AI awareness based on their *p*-value ($>0.05$). These findings demonstrated that the respondents' level of awareness of AI was not dependent on the demographics of the respondents (*p*-value $>0.05$).

## Level of knowledge

Table 4 shows that higher levels of knowledge were statistically significant to the participants who resided in rural areas and worked as farm owners (*p*-value $< 0.05$). Gender, age,

Table 4  Relationship between knowledge level and demographic characteristics.

| Demographics | Knowledge level | | | P-value |
|---|---|---|---|---|
| | High N (%) | Moderate N (%) | Low N (%) | |
| Gender | | | | |
| Male | 53 (44.53) | 31 (26.05) | 35 (29.42) | 0.422 |
| Female | 31 (38.27) | 19 (23.45) | 31 (38.27) | |
| Age | | | | |
| 18 years | 2 (100) | 0 (0) | 0 (0) | 0.277 |
| 20–30 year | 21 (42) | 16 (32) | 13 (26) | |
| 31–50 year | 61 (41.21) | 34 (22.97) | 53 (35.81) | |
| Residence | | | | |
| Urban | 23 (25.84) | 25 (28.08) | 41 (55.05) | <0.0001[*] |
| Rural | 61 (54.95) | 25 (22.52) | 25 (22.52) | |
| Religion | | | | |
| Muslim | 76 (40.64) | 50 (26.73) | 61 (32.62) | 0.384 |
| Christian | 4 (80) | 0 (0) | 1 (20) | |
| Hindu | 1 (50) | 0 (0) | 1 (50) | |
| Catholic | 3 (50) | 0 | 3 (50) | |
| Educational Status | | | | |
| Non-to primary | 1 (20) | 3 (60) | 1 (20) | 0.186 |
| Higher than Primary | 83 (43.91) | 47 (24.10) | 65 (33.33) | |
| Working Status | | | | |
| Farm owner | 46 (56.09) | 15 (18.29) | 21 (25.60) | 0.003[*] |
| Paid employees | 38 (20.87) | 35 (29.66) | 45 (38.13) | |

**Notes.**
[*]$p$-value $< 0.05$ is statistically significant.

religion, and educational status, on the other hand, did not show a significant association with knowledge because their $p$-values were not statistically significant ($>0.05$). Figure 2 depicts the AI knowledge levels.

## DISCUSSION

The goal of this cross-sectional survey was to determine the AI awareness level and identify the factors and sources of information related to knowledge among poultry farmworkers in Indonesia. AI is a zoonotic disease mainly affecting birds and other mammals, including humans. The disease is still endemic in Indonesia (*Rehman et al., 2022a*; *Rehman et al., 2022b*; *Rehman et al., 2022c*; *Pusch & Suarez, 2018*; *Wibawa et al., 2014*). The widespread AI epidemic in domestic birds is a key risk factor as it increases the chances of mutations and genetic re-assortment (*Trampuz et al., 2004*). Most of the respondents had good AI knowledge in terms of infection, transmission, and risk variables according to the results.

Our findings revealed crucial information about the knowledge level of people known to be at high risk of AI infection. More than 60% of the study participants said that they had heard about AI, whereas the majority were aware that AI is a contagious infection that affects all birds. This is an important aspect of AI control as it might be influenced

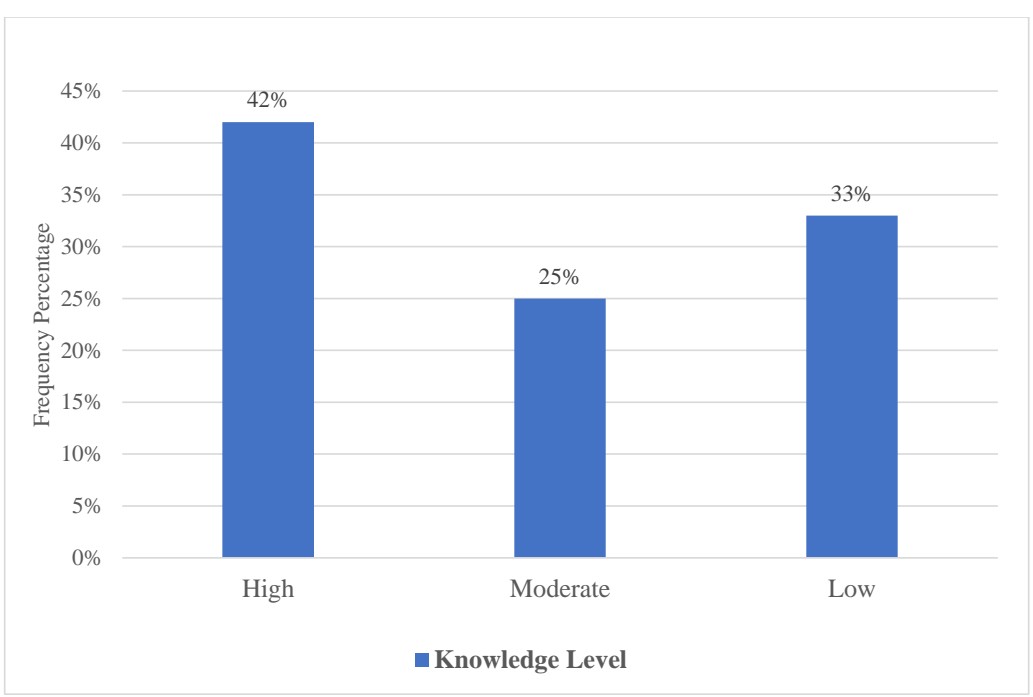

**Figure 2  Graphical representation of knowledge level.**

by the information and experiences gathered from the numerous AI epidemics that have occurred in Indonesia, particularly on Java Island. This study's findings are in line with those of other studies conducted in Ghana and Bangladesh, which showed that 63.5% of the respondents were aware of AI (*Asare et al., 2021*; *Islam et al., 2017*). A previous study conducted in Italy by *Abbate et al. (2006)* found that 64% of poultry workers correctly identified AI as a contagious infection caused by a virus that can affect all species of birds. This study's results also concur with those of a study conducted in Pokhara, Nepal, in which 75% of participants correctly identified avian flu (*Timilsina & Mahat, 2018*). This could be because the authors surveyed a population with a high educational level.

The majority of respondents were aware that AI can be transmitted from animal to animal and from animal to human, according to our findings, whereas only 20% stated that it could be transmitted from human to human. This is in line with previous studies conducted in Italy, India, Nepal, and Bangladesh (*Ezeh, Ezeh & Afolayan, 2017*; *Kumar et al., 2013*; *Lambrou et al., 2020*; *Sarker et al., 2016*). Risk factor assessments revealed that 90%–95% of participants said that poultry and other birds were the AI transmission vehicles, whereas 74.5% and 79.0% believed that veterinarians and poultry workers were high-risk groups of acquiring AI infection. This is similar to findings from a previous study conducted in Baghdad, in which a majority of participants stated that poultry and wild birds were the primary AI transmission vehicles (*Al-Sarray, 2018*). However, this study's results might be higher than the findings of a previous study in Indonesia, which found that only 58.0% of participants believed that diseased birds might transmit HPAI

(*Kurscheid et al., 2015*). Other categories, such as at-risk populations (veterinarians, poultry workers), elicited mixed responses, which contrasted with our findings (*Al-Sarray, 2018*). Direct contact with infected birds had been identified as the primary risk factor for AI transmission among humans in various studies. A cohort study of poultry workers in Hong Kong found that exposure to live poultry increased the AI infection risk among poultry workers and veterinarians (*Bridges et al., 2000*). A previous survey on the knowledge, attitudes, and practices of AI conducted in China revealed a high knowledge level among people living in urban and rural areas, which was in line with our findings. Additionally, this survey found that poultry workers and veterinarians were at a higher risk of contracting AI, which was also consistent with our risk analysis findings (*Xiang et al., 2010*). This study's findings on AI transmission were similar to those reported in a previous study conducted in Indonesia, which revealed that the respondents had a good understanding of AI transmission (*Hunter et al., 2014*). In this survey, it was also clear that the main sources of information for respondents were mass media and health workers, followed by TV and radio. These findings are in line with results of earlier studies conducted among Cambodian and Nigerian poultry workers, where TV and radio were important sources of AI awareness (*Fatiregun & Saani, 2008*; *Khun et al., 2012*). A comparable study conducted in Nepal revealed that TV and newspapers were important sources of campaigns regarding AI knowledge and awareness (*Neupane et al., 2012*). This study's results were consistent with those of *Hunter et al. (2014)* and *Tiongco et al. (2011)*, who said that TV was the main source of AI-related information in Indonesia. In the current investigation, the demographic characteristics did not affect AI awareness, which might be because of the endemicity of AI in Indonesia. In contrast, age, marital status, residency, educational level, and years of job experience had significant impacts on awareness in a previous study in Ghana (*Asare et al., 2021*). Contrastingly, a previous study in Indonesia showed that the level of education had a significant effect on the level of awareness regarding AI (*Tiongco et al., 2011*). According to this survey's findings, 42.0% of respondents had a high level of knowledge, whereas 25.0% had a moderate level of understanding (Fig. 2) of AI illnesses in birds, the source of virus transmission, and other risk categories. In contrast, the study in Ghana indicated that 87.5% of respondents had little understanding of AI pathogenesis, symptoms in diseased birds, and the source of virus transmission (*Asare et al., 2021*). This survey's findings are consistent with the previous H5N1 surveys conducted in China, Laos, and Italy (*Abbate et al., 2006*; *Di Giuseppe et al., 2008*; *Xiang et al., 2010*). A previous study conducted in Indonesia reported that 40% of participants were aware that disposing of diseased dead birds reduced the risk of virus transmission (*Kurscheid et al., 2015*).

Our findings are intended to assist decision-makers in improving AI control and prevention strategies among poultry farmworkers through educational initiatives (workshops, seminars, *etc.*), mass media, health workers, TV, and radio as the main information sources.

## LIMITATIONS OF THIS STUDY

The major limitations in this study were attributed to the sampling method used and the regions covered, as these findings cannot be extrapolated to all of Indonesia. This is

because not enough social resources were available to cover more Indonesian provinces. Furthermore, as this is an online survey, the respondents' interpretations of certain questions were susceptible to variations. Only the sociodemographic factors, knowledge, and awareness were examined as influencing factors to avoid having too many items in the questionnaire, inadvertently causing a long response time. Moreover, this study was feasible only for people who had smartphones, used WhatsApp, had email IDs, and worked on commercial farms. Additional assessments based on all elements of the knowledge related to AI would be necessary to ascertain the true degree of knowledge among local farmworkers. Collecting data was problematic because our survey was conducted online. For the study questionnaire distribution, WhatsApp and email were chosen as our modes of communication. Most respondents are less likely to stay completely engaged for a survey lasting >8–10 min, which caused the low response rate. The respondents were requested to complete the survey questionnaire several times, but the majority did not. Not meeting, the minimum sample size and the consequences for validity and interpretation of the findings are included in the study limitations.

## CONCLUSIONS

This study's findings revealed that poultry workers of the studied areas had good knowledge about the infection, transmission, and risk factors associated with AI. The primary sources of information about AI were health workers and TV. In addition, veterinarians and poultry workers were at a higher risk of contracting AI infection than butchers. Furthermore, farm owners and workers in rural areas were shown to have a better degree of AI knowledge than those in urban areas. However, Indonesian poultry farm employees must further increase their AI knowledge because of the high infection risk. This study's findings may help to improve AI policies and targeted management strategies in controlling and eradicating the disease in Indonesia.

## ACKNOWLEDGEMENTS

We would like to express our gratitude to all the veterinarians, regional veterinary offices, universities, and general public of Indonesia for their assistance in collecting data for this study.

### Funding
This work was supported by the Universitas Airlangga, under project number 418/UN3.15/PT/2021. The funders had no role in study design, data collection and analysis, decision to publish, or preparation of the manuscript.

### Grant Disclosures
The following grant information was disclosed by the authors:
Universitas Airlangga: 418/UN3.15/PT/2021.

## Competing Interests

The authors declare there are no competing interests.

## Author Contributions

- Saifur Rehman conceived and designed the experiments, performed the experiments, analyzed the data, prepared figures and/or tables, authored or reviewed drafts of the article, and approved the final draft.
- Aamir Shehzad analyzed the data, authored or reviewed drafts of the article, and approved the final draft.
- Lisa Dyah Andriyani conceived and designed the experiments, authored or reviewed drafts of the article, and approved the final draft.
- Mustofa Helmi Effendi conceived and designed the experiments, performed the experiments, authored or reviewed drafts of the article, and approved the final draft.
- Zain Ul Abadeen analyzed the data, authored or reviewed drafts of the article, and approved the final draft.
- Muhammad Ilyas Khan conceived and designed the experiments, prepared figures and/or tables, and approved the final draft.
- Muhammad Bilal performed the experiments, prepared figures and/or tables, authored or reviewed drafts of the article, and approved the final draft.

## Data Availability

The data is available at figshare: Rehman, Saifur; Shehzad, Aamir; Andriyani, Lisa Dyah; Helmi effendi, Mustofa; Abideen, Zain Ul; et al. (2022): An Epidemiological Survey of Avian Influenza Knowledge and Practices among Poultry Farm Workers in Indonesia. figshare. Dataset. https://doi.org/10.6084/m9.figshare.19087538.v2.

## Supplemental Information

Supplemental information for this article can be found online at http://dx.doi.org/10.7717/peerj.14600#supplemental-information.

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
