# Peer review of "A cross-sectional survey of avian influenza knowledge among poultry farmworkers in Indonesia"

_PeerJ, doi:10.7717/peerj.14600_

## Round 0.1 · original submission · Major Revisions

Dear Authors,

We are aware of the effort made on this new submission (after the rejection for plagiarism). All reviewers highlighted their interest in the data that are added to the community knowledge about Avian influenza awareness in poultry workers. There are, however, major revisions to address, particularly on the method description, before we can consider this paper suitable for publication in PeerJ.

Best regards

·

Basic reporting

- Minor correction in English grammar is needed.
- Literature references and field background/ context was provided.
- Raw data was shared.
- Self-contained with relevant results to hypotheses was provided.

Experimental design

- Study design was well-defined, relevant, and meaningful to fulfill the knowledge gap.
- Detailed explanation or clarification on materials and method was provided.
- However, scoring for the categories will be questionable and need more clear cutoff points (for example: "sometimes" vs "never".

Validity of the findings

- Impact and novelty are not assessed.
- Meaningful replication where rationale and benefit to literature are clearly stated.
- Discussion and conclusions are well stated linked to the original research question and limitations were described.

Reviewer 2 ·

Basic reporting

The paper has been improved in both writing and study design. However, it still needs to be revised in some places to avoid confusing. For example, L31 in the abstract part, the author wrote “ aged <20-50 years”. it should be written clearly as “ aged from 18 to 50 years”.

Experimental design

It is very good when the author added the criteria for selection of study areas, sample size calculation in the materials and methods part. However, it is not clear when the author mentioned “a big commercial farm” then noted it as “(broiler, breeder, layer, and backyard)” (L140). From my knowledge about farming system, backyard is much different from big commercial farm. Text from L134 to L140 should be revised to be more precise.

Validity of the findings

The paper provides descriptive data on KAP of poultry farm workers in Indonesia. The results part was revised to be short and more concentrated.
The conclusion part should be revised to be succinct. Justifications for the necessary of improving Indonesian poultry farm workers’ knowledge and practice (L342 to L348) should be mentioned in the discussion part.

Additional comments

- Table 3: in P value column, p values were marked with one or two symbol ”*”. The author should explain the significance of this symbol and note it below the table. This symbol in the table 3 will make confuse with those at p values in tables 4, 5. Similarly “95% C.Interval” should be changed as “95% CI” in the table and note the full letters below the table.
- Notes for symbol “*” below table 4 and table 5 should be consistent.
- Table 5: it should change the column title “P-value*” by “P-value” to get the consistence between tables 3,4. Other column titles (ie. High frequency, Moderate, Low) should be added with “%” to be consistent with column titles in table 4.
- Figure 2: the author wrote “p-value=0.009*” but did not note what does the symbol “*” mean. I suggest the author should add notes for p value and this symbol below the figure.

Reviewer 3 ·

Basic reporting

This article presents data on avian influenza information sources and basic concepts of AI transmission for a small sample of poultry workers in Indonesia. Although the manuscript also reports on practices, this data is obtained through a poorly designed questionnaire and as per my comments detailed below, should not be included. I also suggest to rename the title of the study to better reflect the content, e.g., A cross-sectional survey of avian influenza knowledge among poultry farm workers in Indonesia.
The language needs to be improved throughout the manuscript but especially so in the methods and results sections and the use of decimals needs to be consistent.
I have made some comments to abstract section on the PDF (please see annotated PDF). Commetns for all other sections are in the below sections.
I recommend the authors rewrite the introduction. In its current format, it is too long, has no logical flow (the statements jump around and seem out of place) and seems like a mishmash of statements copied from other articles (for e.g., lines 67-71 and 106-108, Kurscheid et al 2015). Consequently, the references are outdated, especially in the context to the wording used by the authors (e.g., lines 57-58 and 61-62). There is frequent mention of ‘pandemic’ which are not relevant to the context of the article (e.g., 71-73, 78-80, 114-116). The statement made in lines 80-81 needs to be expanded (i.e., how did the authors come to this conclusion? Perhaps provide some examples). There are statements that needs references (e.g., lines 84-86, 101-102). Regarding lines 103-104 – monitoring of what exactly? The authors have also neglected to discuss the findings of other similar studies conducted in the country of interest (Indonesia), which would be much more relevant that referring to studies in other countries where the situation among poultry workers could be very different to Indonesia. Subsequently, the introduction does not provide adequate context for the objective of this study.

Experimental design

The notes below refer to the materials and methods section of the manuscript.

After reviewing the methods section and the questionnaire, I have serious concerns about the validity of the data in this manuscript. The questions related to practices are insufficient and too ambiguous to obtain useful data and therefore the results should not be included in this manuscript. For example: separate clothes for what? What kind of contact with bird cages? Use of face mask, boots/boot covers, handwashing when? Consult doctors when and for what? What is defined as proper disposal of dead birds. Was the questionnaire validated? The other sections are also very limited. The questions are far too vague, and the way it is worded, those in the section of vehicles of transmission are no different to those above it. After checking the original questionnaire (Abbate et al 2006), I am curious as to why the authors chose to simplify their questionnaire to such an extent. Also, who did the translation (to Indonesian) and was it back translated to ensure there was no mistranslation?
The English informed consent form is not relevant for this study as it is targeted at students. Please provide a copy of the Bahasa Indonesia version in English as this one is aimed at poultry workers. Also, why was informed consent not listed as a criteria for study participation?
Could the authors please provide the raw data file as an excel document. I was unable to review the file as I do not have or use SPSS. I did find the data available on a data repository site which needs to be referenced in the manuscript.
The methods section also needs to be reworked. Currently, a lot of the content under ‘Study area and population’ should be under separate headings. Please break up into appropriate sections as per standard scientific journal format.
Lines 134-136: what is meant by significant number of commercial farms? How many? And how do those numbers compare to other provinces. More detail is needed here.
Line 136: What is meant by majority? What proportion?
I cannot comment on the sample size determination approach as I am not a statistician but I am curious as to what the authors based their assumption of 20,000 poultry farm workers on.
The statement at lines 156-157 is unclear. Please clarify.
Line 160: please state who ‘those groups’ refers to.
Lines 162-164: what is meant by ‘author online’…?

Validity of the findings

The comments below refer to the results and discussion section of the mansucript.

As mentioned above, the data presented for 'practices' should not be included in this manuscript.
Associations between source of AI information or mode of transmission and demographic variables should be listed under their respective headings not under risk groups and practices. The latter section should be removed as the questionnaire was insufficient to report data on this topic as per my earlier comment.

Lines 258-259: it is better to use original references rather than a fact sheet.
Lines 259-260: Please rephrase this statement. The study is not an epidemiological study of AI. It is a cross-sectional survey of poultry farm workers with regard to sources of AI information and basic knowledge of AI transmission.
As per my comment related to the introduction section, why have the authors not discussed their findings in relation to AI information sources and knowledge from other studies conducted in Indonesia?
Lines 292-293: the use of the term specifically does not make sense here. Please remove.
Lines 307-317: reference to practices should be removed as per my earlier comment.
Limitations/conclusion:
The authors did well to address the many limitations of the study. I can understand that the authors wanted to improve recruitment by simplifying the questionnaire but unfortunately this was done very poorly, which has ultimately limited the importance and relevance of this study further. The small number of participants also meant that few statistically relevant findings could be found.

Additional comments

Based on my assessment of this manuscript, it needs extensive revision before it can be considered for publication. The questionnaire was very poorly designed and at a minimum, the section on practices needs to be removed entirely and only the findings related to sources of AI information and the basic understanding of transmission between birds, animals and people should be presented.

Annotated reviews are not available for download in order to protect the identity of reviewers who chose to remain anonymous.

---

## Round 0.2 · Major Revisions

Following the reviewers' responses, the authors have answered some of the comments and the manuscript has improved. However, there are still some concerns, particularly about the methodology description. Please answer comments from the 3 reviewers and proofread your next submission

·

Basic reporting

Minor correction of English language expression is still needed.
Proper use of punctuation and tense is required.
Literature references with background and context were provided.
Raw data and figures were provided.
It still needs to check the structure (including line spacing, paragraph spacing, and sentences)
Relevant results were provided.
The consistency of decimal places is required.
In addition, it would be great if the author could provide information on how they identify the cutoff point/ value.

Experimental design

Research questions were well defined and relevant; the author also stated how research fills the knowledge gap.
Regarding methods, it would be better if the author could provide information on how they determined cutoff points and the total number of respondents used for data analysis at the end.
Even though the author mentioned that two categories were combined/ merged, did not mention the reason such as statistics or underlying assumptions.
The consistency of decimal places is required across the document.

Validity of the findings

Initially, the author mentioned that the response rate was 46% (210 respondents). In the result section, I noticed that 10 were dropped off due to not completing the questionnaire. So I think it would not be 46% of the total respondents for data analysis but 44.4%. So would you like to keep it as 46% or 44.4%?
From this information, the response rate was <50% and less than the calculated sample size of 377. How would you validate the response would be representative of the whole population? I think it would be better to provide the confidence interval.
The underlying data were provided.
Conclusions and limitations were well stated.

Additional comments

Introduction
• Line 38 – Would it be better to use the word “media” rather than “TV”
• Line 64-65: “Human death cases in developing countries such 65 as China were at 100% in 2003, but have since dropped to 50% in 2010” to be modified to be more relevant. “Example: Human mortality rate in developing countries seems to change over time. In China, ……”
• Line 73: “)” is missing
• Line 75: Putting “full stop” twice
• Line 81 and 84: No “full stop” was noted at the end of the sentence
• Line 85: Please remove “(“ at the beginning of the sentence
• Line 89 and 90: “The data on the importance of knowledge in the context of the influenza outbreak has been less overwhelming.” – What do you mean by this sentence?
• Line 91 and 92: It would be better to change from “Prior study” to “the previous study”
• Line 135: I would suggest changing something like this- “The findings of this study are expected to help policymakers enhance AI knowledge and preventative practices among poultry farm workers through educational initiatives (seminars, workshops).”
Materials and Methods
• Line 143: Please rephrase the sentence “Participants were given verbal information about the study’s aims, purpose, and structure, as well as assurances of confidentiality”
• Line 159: From “The inclusion criteria were based on,….”, remove “based on,”
• Line 162: Please change “respondent accuracy” to “the accuracy of response or the response accuracy”
• Line 178 and 182: It would be better to use “veterinary doctors” than “veterinarians”
• Line 187: “In the end, there were found to 188 be only 10 incomplete questionnaires, and these were removed from the final analysis.” – from this sentence, does it mean out of 210 respondents?
• Line 197: Please remove “full stop” there
• Line 209: does it mean “at risk of contacting AIV” in “……..at risk of contracting AI was used to assess”?
• Line 217: How did you categorize “don’t know”?
• Line 217 and 218: Did you test if there is no significant difference between “sometimes” and “never”? How did you decide to combine/ merge these two categories?
• Line 218: How did you identify the cutoff points? Please explain.
Result
• The consistency of decimal place was required.
• I noticed that 10 were dropped off due to not completing the questionnaire. So I think it would not be 46% of the total respondents for data analysis but 44.4%. So would you like to keep it as 46% or 44.4%?
Discussion
• Line 305: I think you need to add “)” and “full stop” in “…..(Ezeh et al., 2017; Kumar et al., 2013; Lambrou et al., 2020; Sarker et al., 305 2016).”
• Line 311: I think you need to add “full stop” in “….(Kurscheid et al., 2015)”
• Line 355: What do you mean “on the same” in the sentence “……through education initiatives (workshops, seminars, etc.) on the same”.
• In addition, I notice that mass media, health workers, TV, and radio were the main source of information. Apart from workshops, and seminars, would it be effective to promote awareness through mass media as well? I would suggest highlighting it at the end of the discussion (in line 355).

Reviewer 3 ·

Basic reporting

The manuscript has been satisfactorily improved in terms of language, references and data shared.

Experimental design

The authors have not satisfactorily addressed my main concerns regarding the research methods (design and validity of the questionnaire and its findings) and therefore I do not support publication of the manuscript in its current format.

Validity of the findings

Please see my comments above and below.

Additional comments

I thank the authors for taking into consideration the comments from the reviewers. Although sections of the manuscript have been improved and most of my comments have been addressed, there are still some outstanding points. In relation to my previous comments and the authors’ responses, please find my follow up responses below:
• I understand the authors concern that removing the section related to practices would lose its impact but unfortunately, the methods do not scientifically warrant inclusion of the content. A questionnaire is a tool to elicit information but the quality of the data is only as good as the quality of the questions. As I had mentioned in my previous comments, I do not understand why the original questionnaire (Abbate et al 2006) was adapted so significantly. In the current study’s questionnaire, the questions were poorly phrased which meant that the applicability of the responses was limited and not sufficient to draw any meaningful conclusions related to practices to minimise AI transmission. One cannot retrospectively change the questionnaire and present the original results. The authors should carefully consider whether they wish to present the valid findings of the questionnaire, which in this case are those primarily related to sources of information and risk groups or if they wish to revise the questionnaire and conduct the survey again.
• The title of the manuscript remains unchanged. As I mentioned previously, this study is not an epidemiological study of AI. It is a cross-sectional survey of poultry farm workers with regard to sources of AI information and basic knowledge of AI transmission.
• The introduction has improved significantly and the authors have addressed most of my concerns in this section. However, I do not believe that the objectives of the study outlined in the introduction have been adequately addressed in the study.
• All other comments from my earlier feedback have been satisfactorily addressed.

·

Basic reporting

The authors have made efforts to respond to the comments of previous reviewers and I believe that a number of them are at the beginning of their peer-reviewed publishing careers. The work has content but I will encourage an English editor to assist in reading through the document before publication if the Editor accepts the manuscript. Some of the English languages were still ambiguous and constricted/twisted making flow of the manuscript difficult.

Experimental design

Good.

Validity of the findings

Findings were valid but presentation can be improved.

Additional comments

The paper may be considered for acceptance if the queries are addressed and inputs accepted, and if proofread as suggested.

---

## Round 0.3 · Minor Revisions

Improvement in the paper has been noticed by reviewers. Please modify the paper according to the reviewers' comments. Please pay particular attention to the English language and consistency in format in the paper.

·

Basic reporting

- Minor correction of English language expression (including tense, use of punctuation marks) is still needed
- Literature references, sufficient background or context were provided
- The structure of article, raw data, figures and other relevant documents were provided. However, it may still need to recheck the line spacing, use of punctuation marks, connecting words and so on.

Experimental design

- Research questions were well defined and relevant; the author also stated how research fills the knowledge gap.
- Ethical approval was also provided and the authors seem to follow the ethical standard
- Authors provided the detailed information of the methods.

Validity of the findings

- The underlying data was provided
- The results of the study were well described.
- As a suggestion, since the data was not collected from the whole population, it would be better if the authors could provide the upper and lower confidence limits.
- Conclusions based on the findings was provided
- Limitation was well stated.

Additional comments

- The authors addressed most of the previous comments and noted some improvement.
- However, I would suggest the following issues for better improvement:
Line 33: Does it mean a total of 200? Or I would suggest the exact number for each gender
Line 51: "..can present themselves in a variety of ways, depending on their virulence". What does a variety of ways? Clinical signs or severity?
Line 58: Please check the tense
Line 84-85: "The data...." What do you mean there? What kind of data? I would suggest to add some connecting words to link the next sentence or add some references.
Line 92: I don't understand the sentence "poultry farmers are the most susceptible..". Do you mean other people (e,g, veterinarians) are not susceptible to AI?
Line 111: "The poultry workers and traders are not involved in disease control...." Does it mean there is no stakeholder involvement or participation when government implement restriction of animal movement, vaccination, etc.?
Line 125: I think from the findings of this study, we could only assess and could not "determine the level....."
Line 157: "The aim of the study.......before they filled out the study questionnaire" I am not sure about the sentence. Is it for participant's consent or the aim of the research. Please rephrase or rewrite
Line 170 and 174: I would suggest to write "veterinarians" instead of "veterinary doctors"
Line 173: You mentioned that "small-scale production farms were excluded". In the materials and methods section, you indicated that you collected data from backyard farms as well. How do you define the backyard farms, small and large scale farms. Do you mean backyard farms raising a large number of poultry? In this case, how would you define small scale?
Line 262: "The pan zoonosis of.....AI in domestic birds is a key risk factor,.." Could you please rephrase or rewrite to avoid confusion?
Line 269: "more than sixty percent said that they had heard of it." Does it mean they know the causal agent, risk or just know the name of disease?
Line 288: "However, our result was higher than the findings..." I would suggest it would be risky to conclude this way as the data was not collected from the whole population and the confidence limits were not provided to compare the two studies.
Line 529: I would suggest to write the reference in appropriate format
Table 3: I would suggest to check the table format to be presentable.
Table 4: Notation need to be clearly described in "*= p-value <0.05 is statistically significant"

Reviewer 3 ·

Basic reporting

See below and attached annotated PDF for summary.

Experimental design

See below and attached annotated PDF for summary.

Validity of the findings

See below and attached annotated PDF for summary.

Additional comments

- The Author’s have changed the title and removed the sections related to practices as per my recommendation and there has been an overall improvement but there are still some minor comments that need to be addressed before the manuscript is ready for publication. Please the annotated PDF and below for further comments.
- Reviewer #1’s comment regarding sample size not meeting the minimum (of 377) and how this impacts the findings has not been appropriately addressed.
- Language still needs further editing, especially in the methods and results sections. See the annotated PDF for examples. Would strongly suggest the Authors seek the services of a professional English-speaking scientific editor.
- Line 151, the Authors’ state ‘big’ commercial farms but they do not provide a definition. Do they mean ‘large-scale’ commercial poultry farms? I suggest looking into FAO definitions.
- There is still inconsistency in use of decimals in the results. Please go through the whole manuscript line by line and check for consistency. I would suggest using 1 decimal place based on what has been reported but it needs to be used for all even if the number is 23.0% for example. The only place this is not necessary is when reporting the findings from other published papers.

Annotated reviews are not available for download in order to protect the identity of reviewers who chose to remain anonymous.

·

Basic reporting

The basic reporting has increased significantly.

Experimental design

The experimental design has been adjusted as suggested. The content was there originally but poorly written in some respect.

Validity of the findings

The data presented, the result and the discussion now met all the criteria outlined on the side bar.

---

## Round 0.4 · accepted · Accept

The last round of revisions reached to minor decision. I reviewed by myself the last changes. and reviewers' comments were addressed